# WSDMS: Debunk Fake News via Weakly Supervised Detection of Misinforming Sentences with Contextualized Social Wisdom

**Ruichao Yang**[1], **Wei Gao**[2], **Jing Ma**[1,*] **Hongzhan Lin**[1], **Zhiwei Yang**[3]

[1]Hong Kong Baptist University, Hong Kong SAR, China
[2]Singapore Management University, Singapore
[3]Jinan University, Guangzhou, Guangdong, China
{csrcyang,majing,cshzlin}@comp.hkbu.edu.hk,
weigao@smu.edu.sg, yangzw@jnu.edu.cn

## Abstract

In recent years, we witness the explosion of false and unconfirmed information (i.e., rumors) that went viral on social media and shocked the public. Rumors can trigger versatile, mostly controversial stance expressions among social media users. Rumor verification and stance detection are different yet relevant tasks. Fake news debunking primarily focuses on determining the truthfulness of news articles, which oversimplifies the issue as fake news often combines elements of both truth and falsehood. Thus, it becomes crucial to identify specific instances of misinformation within the articles. In this research, we investigate a novel task in the field of fake news debunking, which involves detecting sentence-level misinformation. One of the major challenges in this task is the absence of a training dataset with sentence-level annotations regarding veracity. Inspired by the Multiple Instance Learning (MIL) approach, we propose a model called Weakly Supervised Detection of Misinforming Sentences (WSDMS). This model only requires bag-level labels for training but is capable of inferring both sentence-level misinformation and article-level veracity, aided by relevant social media conversations that are attentively contextualized with news sentences. We evaluate WSDMS on three real-world benchmarks and demonstrate that it outperforms existing state-of-the-art baselines in debunking fake news at both the sentence and article levels.

## 1 Introduction

Misinformation, such as fake news, poses tremendous risks and threats to contemporary society. The detection of fake news entails various technical challenges (Glockner et al., 2022), and one of them is accurately identifying false elements within news articles. This challenge arises due to the blending of authentic and fabricated content by creators of fake news, thereby complicating the determination of overall veracity (Solovev and Pröllochs, 2022). Such instances have been prevalent during the Covid-19 pandemic[1].

Fake news detection aims to determine the veracity of a given news article (Shu et al., 2017). Previous analysis has revealed that users often share comments and provide evidence about fake news on social media platforms (Zubiaga et al., 2017), which has led to a growing stream of research that leverages these social engagements, along with the content of news articles, to aid in fake news detection (Pan et al., 2018; Shu et al., 2019a; Min et al., 2022). This approach bears analogies to rumor detection, where the focus is on assessing as a specific statement rather than an entire news article (Wu et al., 2015; Ma et al., 2018; Bian et al., 2020; Lin et al., 2021; Song et al., 2021; Park et al., 2021; Zheng et al., 2022; Xu et al., 2022). Many studies in this domain aims to train supervised classifiers using features extracted from the social context and the content of the claim or article. However, the existing fake news detection models predominately focus on coarse-level classification of the entire article, which oversimplifies the problem. Misinformation can be strategically embedded within an article by manipulating portions of its content to enhance its credibility (Feng et al., 2012; Rogers et al., 2017; Zhu et al., 2022) Therefore, we target a fine-grained task that aims to identify sentences containing misinformation within an article, which can be jointly learned with article-level fake news detection.

Figure 1 shows an illustrative example of a fake news article titled "NASA will pay 100,000 USD

---

*Jing Ma is the corresponding author.

[1]https://ahmedabadmirror.com/gujarat-plans-to-give-world-a-wonder-drug-to-battle-corona/76017951.html. This article combines factual information about the historical use of cow urine in India's traditional medicine with false assertions that cow urine contains active ingredients capable of treating Covid-19 and has been used in hospitals in South Korea and China.

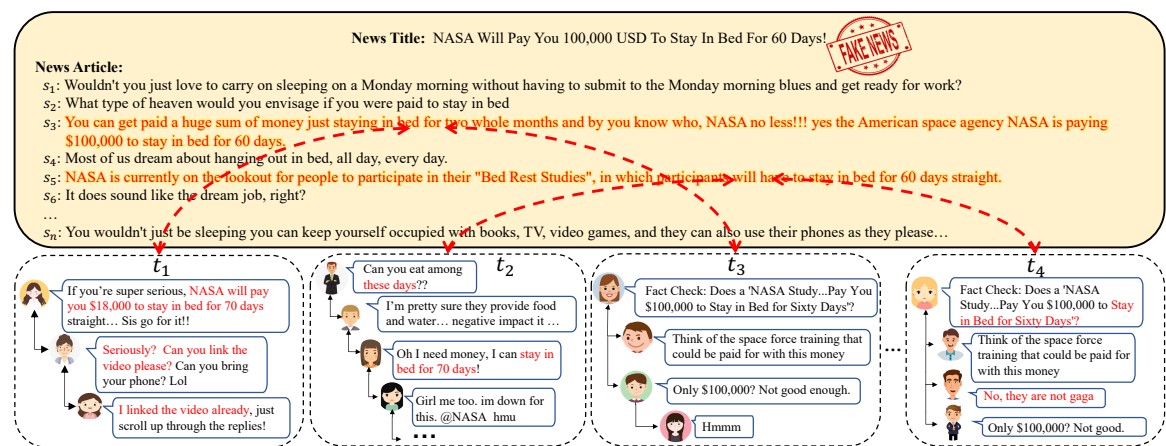

Figure 1: A fake news article together with its relevant social context information, where the sentences containing misinformation (i.e., $s_3$ and $s_5$) are in orange and the posts implying the misinforming sentences are in red.

to participants staying in bed for 60 days!", where the sentences in the article can be linked to a set of social conversations organized as propagation trees of posts. These sentences contain opinions and evidence that can aid in the veracity classification at the sentence and article levels, specifically in spotting misinformation sentences. For instance, sentence $s_3$ can be debunked by referring to trees $t_1$ and $t_3$, as they provide evidence that contradicts the incorrect reward amount and duration mentioned in the sentence. This information helps in determining that the article is fake. Conversely, if we already know that the article is fake, we can infer that there must be misinforming sentences present within it.

However, existing methods are not readily applicable for the identification of sentence-level misinformation due to two main reasons: 1) Obtaining veracity labels for sentences in an article is costly, as it requires annotators to exhaustively fact-check each sentence. 2) While rumor detection models can predict the label of a given claim, they often assume the availability of social conversations that correspond to the claim. However, it is difficult to establish a correspondence between social conversations and specific sentences within a news article. Inspired by multiple instance learning (MIL) (Foulds and Frank, 2010), we attempt to develop an approach for debunking fake news via weakly supervised detection of misinforming sentences (i.e., instances), called WSDMS[2], only using available article-level veracity annotations (i.e., bag-level labels) and a handful of social conversations related to the news.

To gather the relevant social conversations associated with an article, we employ established

methods used in fake news detection that rely on social news engagement data collection (Shu et al., 2020), which provides the necessary conversation trees linked to the article in question. We devise a hierarchical embedding model to establish connections between each sentence in the article and its corresponding conversations, facilitating the identification of sentence-level misinformation. Standard MIL determines the bag-level label as positive if one or more instances within the bag are positive, and negative otherwise (Dietterich et al., 1997). To improve its tolerance on sentence-level prediction errors, we further develop a collective attention mechanism for a more accurate article veracity inference on top of the sentence-level predictions. The entire framework is trained end-to-end by optimizing a loss function that aims to alleviate prediction bias by considering both sentence- and article-level consistencies. Our approach ensures that the model captures the nuances of misinformation at both levels of granularity. Our contributions are summarized as follows:

- Unlike existing fake news detection approaches, we introduce a new task that is focused on spotting misinforming sentences in news articles while simultaneously detecting article-level fake news.

- We develop WSDMS, a MIL-based model, to contextualize news sentences aided by social conversations about the news and use only article veracity annotations to weakly supervise sentence representation and model training.

- Our method achieves superior performance over state-of-the-art baselines on sentence- and article-level misinformation detection.

[2]https://github.com/HKBUNLP/WSDMS-EMNLP2023

## 2 Related Work

Early studies on fake news detection have attempted to exploit various approaches to extract features from news content and social context information, including linguistic features (Potthast et al., 2018; Azevedo et al., 2021), visual clues (Jin et al., 2016), temporal traits (Kwon et al., 2013; Ma et al., 2015), user behaviors and profiles (Castillo et al., 2011; Ruchansky et al., 2017; Shu et al., 2019b). Subsequent studies have employed neural networks to automatically learn deep feature representations from similar sources of data (Ma et al., 2016; Popat et al., 2018; Ma et al., 2019; Nguyen et al., 2020; Kaliyar et al., 2021; Sheng et al., 2022). Furthermore, researchers have incorporated external knowledge sources(Pan et al., 2018; Dun et al., 2021; Hu et al., 2021) and combined multi-modal data (Wang et al., 2018, 2021; Fung et al., 2021; Wu et al., 2021; Silva et al., 2021; Chen et al., 2022) to enhance learning and improve fake news detection performance. Notably, social context information has played a crucial role in debunking fake news and rumors (Yuan et al., 2019; Khoo et al., 2020; Yang et al., 2022a; Ma et al., 2020; Mehta et al., 2022). The utilization of social context structures has spurred the development of Graph Neural Networks (GNNs) such as Kernel Graph Attention Networks (KGAT) (Liu et al., 2020) and Graph-aware Co-Attention Networks (GCAN) (Lu and Li, 2020), which have demonstrated effectiveness in various fake news-related tasks. However, existing approaches (Shu et al., 2019a; Jin et al., 2022; Yang et al., 2022b) generally aim to detect article-level fake news, which lack the capability to tell which specific sentences contain misinformation.

MIL is a weakly supervised approach that infers instance-level labels (e.g., sentence or pixel) when training data is annotated with bag-level labels (e.g., document or image) (Dietterich et al., 1997). Several MIL variants have been developed based on threshold-based MIL assumption (Foulds and Frank, 2010) and weighted collective MIL assumption (Pappas and Popescu-Belis, 2017), successfully applied in various downstream tasks such as recommendation systems (Lin et al., 2020) sentiment analysis (Angelidis and Lapata, 2018), keywords extraction (Wang et al., 2016), community question answering (Chen et al., 2017), and more recently joint detection of stances and rumors (Yang et al., 2022a). We adopt the weighted collective MIL assumption (Pappas and Popescu-Belis, 2017) to incorporate a weight function over the sentence space to calculate the article veracity probability. This assumption allows us to achieve a more robust prediction, as it avoids bias introduced by less important instances.

## 3 Problem Definition

We define a fake news dataset as a set of news articles $\{\mathcal{A}\}$, where each article consists of a set of $n$ sentences $\mathcal{A} = \{s_i\}_{i=1}^n$ and $s_i$ is the $i$-th sentence. For each article, we assume there is a set of $m$ social conversation trees relevant to it denoted as $\mathcal{T} = \{t_j\}_{j=1}^m$, where $t_j$ is the $j$-th conversation tree containing posts (i.e., nodes) and message propagation paths (i.e., edges) which can provide the social context information for $\mathcal{A}$. Our task is to predict the veracity of information at both sentence level and article level in a unified model:

- **Sentence-level Veracity Prediction** aims to determine whether each $s_i \in \mathcal{A}$ is a misinforming sentence or not given its relevant social context information $\mathcal{T}$. That is to learn a function $f(\mathcal{A}) : s_1, s_2, \ldots, s_n \to p_1, p_2, \cdots p_n$, where $p_i$ is the sentence-level prediction probability as to whether $s_i$ is misinforming or not.

- **Article-level Veracity Prediction** aims to classify the veracity of the article $\mathcal{A}$ on top of the sentence-level misinformation detection. That is to learn a function $g(\mathcal{A}) : p_1, p_2, \cdots p_n \to \hat{y}$, where $\hat{y}$ denotes the prediction as to whether $\mathcal{A}$ is fake or true. Note that we have only article-level ground truth for model training.

## 4 WSDMS: Our MIL-based Model

Detecting more nuanced instances of misinformation at the sentence level solely based on article content is challenging (Feng et al., 2012). Previous studies have demonstrated that social media posts contain valuable opinions, conjectures, and evidence that can be leveraged to debunk claim-level misinformation, such as rumors (Ma et al., 2017, 2018; Wu et al., 2019), where claims, typically presented as short sentences, share similar characteristics with sentences in news articles. We hypothesize that the detection of misinforming sentences can be done by incorporating relevant information from social context associated with the article. We try to establish connections between social conversations and specific news sentences in the article, enabling the contextualization of social

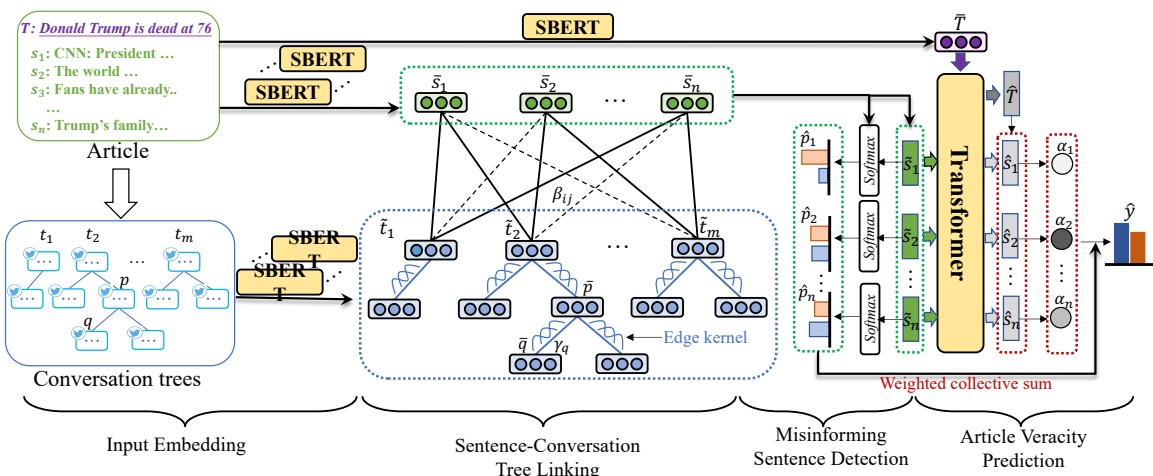

Figure 2: The architecture of our WSDMS model. $\bar{t}_i$ denotes the representation of tree $t_i$ after kernel-based interaction of post information among tree nodes.

wisdom to enrich the representation of sentences, in order to better capture the veracity of sentences.

The architecture of our MIL-based weakly supervised model WSDMS is illustrated in Figure 2. WSDMS consists of four closely coupled components: input embedding, sentence and conversation tree linking, misinforming sentence detection, and article veracity prediction. We describe them with detail in this section.

## 4.1 Input Embeddings

We represent the word sequence of each news sentence and social post using SBERT (Reimers and Gurevych, 2019) which maps the sequence into a fixed-size vector. Let a sequence $S = w_1 w_2 \cdots w_{|S|}$ consist of $|S|$ tokens, where $S$ could optionally denote a news title, a news sentence, or a post in conversation tree. Then, the SBERT embedding of $S$ can be represented by $\bar{S} = \text{SBERT}(w_1, \cdots, w_{|S|})$. In the rest of the paper, given an article $\mathcal{A}$, we will use additional notations $T$ to denote the news title, $p$ and $q$ to denote posts in a conversation tree. And then $\bar{T}$, $\bar{s}_i$, $\bar{p}$ and $\bar{q}$ will denote the respective SBERT embeddings of $T$, $s_i$, $p$ and $q$.

## 4.2 Linking Sentences to Conversation Trees

To mine the discernible relationship between sentences and social posts trees, we want to design a sentence-tree linking mechanism between the sentence set $\{s_i\}_{i=1}^n$ and post tree set $\{t_j\}_{j=1}^m$, both associated with $\mathcal{A}$. There are clearly different designs to create links across the elements between them, such as 1) using a fully connected graph that links any $s_i$ and $t_j$ regardless of their similarity,

followed by a model to fix the closeness of each connection; 2) creating a link according to the similarity between $s_i$ and $t_j$ based on a threshold. Our preliminary experiments indicate that the different designs of interaction indeed influence the performance. Given that the number of sentences and trees associated with articles varies significantly, we opt for the threshold-based approach to avoid the overhead of computing on a fully connected graph. We begin with modeling posts interaction in each tree to learn its representation before linking the sentences and trees.

**Post Interaction Embedding:** To represent a tree accurately, we use a generic kernel-based graph model KernelGAT (Liu et al., 2020) to measure the importance of each post in a tree by modeling the interactions between each post and its neighboring posts.

We first construct a translation matrix $\mathcal{M}$ to represent the similarity of each post with its neighbors, where each $M_{pq} \in \mathcal{M}$ is the cosine similarity between post $p$ and $q$:

$$M_{pq} = \begin{cases} \frac{\bar{p} \cdot \bar{q}}{|\bar{p}||\bar{q}|} & \text{if } q \in \mathcal{N}(p) \\ 0 & \text{otherwise} \end{cases} \quad (1)$$

where $\mathcal{N}(p)$ is the set of neighboring nodes of $p$.

We then define a kernel function $\vec{\mathcal{G}}(M_p)$ to represent the features considering the interactions between $p$ and its neighbors based on $K$ Gaussian kernels (Keerthi and Lin, 2003), and this yields:

$$\vec{\mathcal{G}}(M_p) = \{\mathcal{G}_1(M_p), \cdots, \mathcal{G}_K(M_p)\} \quad (2)$$

where

$$\mathcal{G}_k(M_p) = \log \sum_{q \in \mathcal{N}(p)} \exp\left(-\frac{(M_{pq} - \mu_k)^2}{2\sigma_k^2}\right)$$

and $\mu_k$ and $\sigma_k$ are parameters in the $k$-th kernel to capture the node interactions at different levels (Xiong et al., 2017). Note that if $\sigma_k \to \infty$, the kernel function degenerates to the mean pooling.

Then, we update the representation $\tilde{p}$ of each post $p$ by considering all its neighbors with their identified importance, which is given as:

$$\gamma_q = softmax\left(W_1\left(\vec{\mathcal{G}}(M_p)\right) + b_1\right)[q]$$
$$\tilde{p} = \sum_{q \in \mathcal{N}(p)} \gamma_q \cdot \tilde{q} \tag{3}$$

where $\gamma_q$ is a scalar representing the post-level attention coefficient between $p$ and its neighbor $q$, $W_1$ and $b_1$ are trainable parameters used to transform $K$ kernels into a vector of all nodes in the tree, $[q]$ takes the value corresponding to post $q$, and $\tilde{p}$ and $\tilde{q}$ are initialized respectively with the BERT-based post embeddings $\bar{p}$ and $\bar{q}$.

**Link Sentences and Trees.** With the obtained interaction-enhanced post representations, we use a mean pooling function to represent a conversation tree $t_j$, i.e., $\tilde{t}_j = mean(\sum_p \tilde{p})$ for all $p \in t_j$. For each pair of sentences and tree $(s_i, t_j)$ associated with an article, we then create a link between them if the cosine similarity of $\bar{s}_i$ and $\tilde{t}_j$ is above a global threshold $\tau$, where $\tau$ is determined according to the global range of similarity scores between sentences and trees by mapping $\tau$ to the median of the range of scores. We fix this setting empirically.

### 4.3 Detecting Misinforming Sentences

To spot misinforming sentences based on the graph with the sentence-tree links, we propose a graph attention model to detect whether a sentence $s_i$ contains misinformation. Each sentence can be linked to multiple conversation trees and vice versa. In Figure 1, for example, two trees $t_1$ and $t_3$ are linked to $s_3$, where $t_1$ provides more specific evidence (e.g., the right reward amount and the number of experimental days) indicating that $s_3$ is misinforming, while $t_3$ just implies the sentence is not credible without providing specific clues. Hence, we design an attention mechanism to update the representation of each sentence by considering the importance of all its corresponding trees.

More specifically, let $\mathcal{T}_i$ denote the set of trees linked to $s_i$. We aggregate the representation of corresponding trees according to their attention weights, and then update the sentence representa-

tion. This is achieved by:

$$\beta_{i,j} = \frac{\exp(\tilde{t}_j \cdot \bar{s}_i)}{\sum_{t'_j \in \mathcal{T}_i} \exp(\tilde{t}'_j \cdot \bar{s}_i)}$$
$$\tilde{s}_i = \left(\sum_{t_j \in \mathcal{T}_i} \beta_{i,j} \cdot \tilde{t}_j\right) \oplus \bar{s}_i \tag{4}$$

where $\tilde{s}_i$ denotes the socially contextualized representation of $s_i$, $\beta_{i,j}$ is the importance of $t_j \in \mathcal{T}_i$ with respect to $s_i$, and $\oplus$ denotes concatenation operation.

We then use a fully-connected softmax layer to predict the probability of $s_i$ containing misinformation based on its BERT-based embedding $\bar{s}_i$ and socially contextualized embedding $\tilde{s}_i$:

$$\hat{p}_i = softmax(W_2\tilde{s}_i + W_3\bar{s}_i + b_2) \tag{5}$$

where $W_2$, $W_3$ and $b_2$ are trainable parameters and $\hat{p}_i$ is the class probability distribution of $s_i$ provided that the bag-level class labels are fake and real, based on the MIL (Foulds and Frank, 2010; Angelidis and Lapata, 2018).

### 4.4 Inferring Article Veracity

We can simply predict an article as fake if there is at least one misinforming sentence is detected, which conforms to the original threshold-based MIL assumption. However, the assumption is overly strong because there can be inaccuracies in sentence-level prediction. Based on the weighted collective MIL assumptions (Foulds and Frank, 2010), we design a context-based attention mechanism to bridge the inconsistency between sentence- and article-level predictions.

Specifically, we first learn a global representation for the article utilizing a pre-trained transformer (Grail et al., 2021):

$$[\hat{T}, \hat{s}_1, \cdots, \hat{s}_n] = Trans\left([\bar{T}, \tilde{s}_1, \cdots, \tilde{s}_n]\right) \tag{6}$$

where $\bar{T}$ is the initial SBERT embedding of the article title. We then adopt an attention mechanism to measure the importance of sentences w.r.t the article veracity prediction, which yields:

$$\alpha_i = \frac{\exp(\hat{s}_i \cdot \hat{T})}{\sum_{i=1}^n \exp(\hat{s}_i \cdot \hat{T})}$$
$$\hat{y} = \sum_{i=1}^n \alpha_i \cdot \hat{p}_i \tag{7}$$

where $\alpha_i$ denotes the attention weight of $\hat{s}_i$ relative to the title representation $\hat{T}$, and $\hat{y}$ is the class probability distribution of $\mathcal{A}$ being fake or real.

## 4.5 Model Training

Intuitively, the more similar two sentences are, the more similar their corresponding predictions should be. We define the following loss function considering pairwise consistency between sentence representation and prediction, with only article-level ground truth:

$$\mathcal{L}(\mathcal{A}) = \lambda \cdot \mathcal{C}(\mathcal{A}) + (1 - \lambda) \cdot ||y_\mathcal{A} - \hat{y}_\mathcal{A}||_2^2 \quad (8)$$

where

$$\mathcal{C}(\mathcal{A}) = \sum_{i=1}^{n} \sum_{j=1}^{n} \exp\left(-||\hat{s}_i - \hat{s}_j||_2^2 \cdot ||\hat{p}_i - \hat{p}_j||_2^2\right)$$

Here $\mathcal{C}(.) \in [0, 1]$ is the function measuring the consistency between pairwise sentence similarity (i.e., $\hat{s}_i$ and $\hat{s}_j$) and the prediction (i.e., $\hat{p}_i$ and $\hat{p}_j$), $y_\mathcal{A}$ and $\hat{y}_\mathcal{A}$ denote respectively the ground-truth and predicted class probability distributions of $\mathcal{A}$, $||.||_2^2$ is an efficient kernel based on the L2 norm (Luo et al., 2016) as a non-negative penalty function, and $\lambda$ is the trade-off coefficient.

## 5 Experiments and Results

### 5.1 Datasets and Setup

We employ two public real-world datasets Politi-Fact and GossipCop (Shu et al., 2020) respectively related to politics and entertainment fake news, where relevant social conversations are collected from Twitter. We also construct an open-domain fake news dataset BuzzNews by extending Buz-zFeed (Tandoc Jr, 2018), for which we gather social conversations of the articles via Twitter API[3].

We recruit three annotators to label misinform-ing sentences of the articles in the test sets of the three datasets. We train the annotators by providing them with a unified set of annotation rules referring to the detailed guide from several fact-checking websites such as snopes.com and politifact.com, where specific rationales on how each claim was judged are provided. Then, we take a majority vote for determining the label of each sentence, and the inter-annotator agreement is 0.793. Table 1 shows the statistics of these three datasets.

We use precision (Pre), recall (Rec), F1, and accuracy (Acc) as evaluation metrics. All the baselines and our methods are implemented with Py-Torch (Paszke et al., 2019) (see Appendix A.2 for implementation details).

| | Stat. | PolitiFact | GossipCop | BuzzNews |
|---|---|---|---|---|
| **Train** | # True | 624 | 16,658 | 301 |
| | # Fake | 432 | 5,255 | 105 |
| **Test** | # True | 140 | 160 | 50 |
| | # Fake | 70 | 80 | 25 |
| **Total** | – | 1,270 | 22,153 | 481 |
| | # avg. sent/art | 30 | 27 | 27 |
| | # avg. trees/art | 13 | 16 | 9 |
| | # avg. posts/tree | 316 | 58 | 340 |

Table 1: Statistics of the datasets used.

### 5.2 Article-level Fake News Detection

We compare the following models at the article level. Some original settings of baselines might not suit the data in this task, which have to be specifically customized (see Appendix A.1). 1) **DeClarE** (Popat et al., 2018): An evidence-aware network using news title to attend over words in relevant posts for verifying news claims. 2) **HAN** (Ma et al., 2019): A hierarchical attention network using the news title to attend over relevant posts as evidences. 3) **dEFEND** (Shu et al., 2019a): A sentence-post co-attention network for fake news detection. 4) **BerTweet** (Nguyen et al., 2020): A language model pre-trained on 850M tweets, which is applied here for article verification using article and relevant posts. 5) **GCAN** (Lu and Li, 2020): A graph-aware co-attention model trained on user profile and post propagation structure without using post content to verify the news given title. 6) **Bi-GCN** (Bian et al., 2020): A bi-directional graph convolutional network using news title and propagation structure of posts for verifying the news. 7) **KAN** (Dun et al., 2021): An attention network utilizing entities in article content and entity contexts for fake news detection. 8) **SureFact** (Yang et al., 2022b): A reinforcement subgraph reasoning method using the topic connection between article and relevant posts for fake news detection. 9) **WS-DMS**: Our proposed weakly supervised method. 10) **WSDMS-FC**: A variant of our method that fully connects sentences and post trees. Table 2 presents the following observations:

- In the first group of structured models, dEFEND performs the best. This is because DeClearE and HAN are designed to only use the *external* relevant context of a claim and BerTweet is trained to represent social posts. dEFEND leverages features extracted from both article content and external posts that are complementary.

---

[3]https://developer.twitter.com/en/docs

| Dataset | PolitiFact | | | | GossipCop | | | | BuzzNews | | | |
|---|---|---|---|---|---|---|---|---|---|---|---|---|
| **Method** | **Pre** | **Rec** | **F1** | **Acc** | **Pre** | **Rec** | **F1** | **Acc** | **Pre** | **Rec** | **F1** | **Acc** |
| DeClarE | 0.714 | 0.746 | 0.730 | 0.789 | 0.706 | 0.741 | 0.723 | 0.762 | 0.705 | 0.743 | 0.724 | 0.754 |
| HAN | 0.752 | 0.779 | 0.765 | 0.803 | 0.718 | 0.739 | 0.728 | 0.789 | 0.727 | 0.768 | 0.747 | 0.762 |
| dEFEND | 0.900 | 0.926 | 0.913 | 0.886 | 0.729 | 0.785 | 0.756 | 0.808 | 0.731 | 0.792 | 0.760 | 0.810 |
| BerTweet | 0.844 | 0.903 | 0.873 | 0.878 | 0.851 | 0.862 | 0.857 | 0.848 | 0.831 | 0.840 | 0.835 | 0.811 |
| GCAN | 0.817 | 0.821 | 0.819 | 0.837 | 0.782 | 0.803 | 0.792 | 0.791 | 0.780 | 0.800 | 0.790 | 0.795 |
| Bi-GCN | 0.852 | 0.838 | 0.845 | 0.865 | 0.797 | 0.813 | 0.805 | 0.822 | 0.791 | 0.814 | 0.802 | 0.817 |
| KAN | 0.870 | 0.840 | 0.855 | 0.859 | 0.776 | 0.770 | 0.773 | 0.807 | 0.766 | 0.790 | 0.778 | 0.820 |
| SureFact | 0.913 | 0.939 | 0.924 | 0.887 | 0.859 | 0.872 | 0.865 | 0.847 | 0.841 | 0.856 | 0.848 | 0.829 |
| **WSDMS** | 0.921 | **0.967** | 0.943 | 0.904 | **0.864** | 0.876 | 0.870 | 0.850 | **0.850** | **0.857** | **0.853** | **0.858** |
| **WSDMS-FC** | **0.923** | **0.967** | **0.944** | **0.908** | 0.862 | **0.879** | **0.871** | **0.853** | **0.850** | **0.857** | **0.853** | **0.858** |

Table 2: Article-level fake news detection results.

- In the second group of non-structured models, the graph-based models GCAN and Bi-GCN mainly rely on propagation structures of fake news and perform comparably with KAN using entities and their contexts extracted from the social media content, suggesting that social conversations embed a good amount of human wisdom useful for detecting fake news. SureFact performs best among all the baselines because it groups social posts into the topics discovered from article content, suggesting that creating a connection between them at the topic level is helpful.

- WSDMS consistently defeats the best baseline SureFact on the three datasets, demonstrating that our explicit and fine-grained linking between sentence and social context is superior, and the sentence-level detection can help article veracity prediction. In addition, WSDMS does not sacrifice its performance compared to WSDMS-FC that uses full connections between sentences and trees, while we find that WSDMS significantly reduces training time from 4.5 to 2 hours. This indicates our sentence-tree linking method is cost-effective.

## 5.3 Misinforming Sentence Detection

For misinforming sentence detection, the baselines are deployed by treating each sentence as a claim and the conversation trees linked to the sentence (see Section 4.2) as the source of evidence. SureFact is excluded as it cannot classify specific sentences. More details are in Appendix A.1.

Since all baselines are supervised methods that need sentence labels for training, we split the three test sets with sentence-level annotation into train and test parts with a 70%-30% ratio. Due to the large number of sentences in the original test sets (6,300/6,480/2,480), we end up with three workable sentence-level training and test sets. We then train all models on the same training data. But this intentionally disadvantages our WSDMS since it can only use article labels. Therefore, we also present the performance of WSDMS (o) trained on the original training sets without sentence labels, which baselines cannot take advantage of. Table 3 conveys the following findings:

- Similar to article-level prediction, dEFEND outperforms DeClarE and HAN because it effectively models the sentence and social context correlations via the co-attention mechanism. BERTweet is more advantageous at representing social media posts, demonstrating better performance at the sentence level.

- Among the structured models, KAN performs best because it incorporates both content and propagation information and has a co-attention mechanism between sentence and entity contexts extracted from social conversations. This may enhance sentence representation better than Bi-GCN and GCAN that can only utilize propagation-based features.

- Weakly supervised WSDMS performs better than DeClarE and comparably with HAN, which are fully supervised. This is because WSDMS considers the propagation structure while DeClarE and HAN can only leverage unstructured posts. The overall performance of WSDMS is clearly compromised due to weak supervision. However, when it is trained on the original datasets, WSDMS (o) can enjoy the large volume of article labels to beat all baselines that cannot be weakly supervised. To reach the same level of performance, the baselines may need tremendous sentence annotations which are infeasible to get. Again, it performs comparably well as WSDMS-FC (o), implying that our sentence-tree linking

| Dataset | PolitiFact | | | | GossipCop | | | | BuzzNews | | | |
|---|---|---|---|---|---|---|---|---|---|---|---|---|
| **Method** | **Pre** | **Rec** | **F1** | **Acc** | **Pre** | **Rec** | **F1** | **Acc** | **Pre** | **Rec** | **F1** | **Acc** |
| DeClarE | 0.504 | 0.531 | 0.517 | 0.559 | 0.501 | 0.528 | 0.514 | 0.550 | 0.513 | 0.520 | 0.516 | 0.540 |
| HAN | 0.531 | 0.559 | 0.545 | 0.565 | 0.510 | 0.529 | 0.519 | 0.561 | 0.518 | 0.537 | 0.527 | 0.562 |
| dEFEND | 0.539 | 0.586 | 0.562 | 0.605 | 0.534 | 0.581 | 0.557 | 0.600 | 0.538 | 0.570 | 0.554 | 0.580 |
| BerTweet | 0.542 | 0.630 | 0.583 | 0.619 | 0.539 | 0.619 | 0.576 | 0.602 | 0.542 | 0.610 | 0.574 | 0.599 |
| GCAN | 0.533 | 0.563 | 0.548 | 0.589 | 0.511 | 0.561 | 0.535 | 0.581 | 0.521 | 0.551 | 0.536 | 0.580 |
| Bi-GCN | 0.557 | 0.589 | 0.573 | 0.606 | 0.531 | 0.560 | 0.545 | 0.593 | 0.533 | 0.553 | 0.543 | 0.601 |
| KAN | 0.574 | 0.594 | 0.584 | 0.611 | 0.539 | 0.561 | 0.550 | 0.609 | 0.540 | 0.560 | 0.550 | 0.610 |
| WSDMS | 0.518 | 0.539 | 0.527 | 0.564 | 0.508 | 0.531 | 0.519 | 0.562 | 0.513 | 0.537 | 0.524 | 0.549 |
| **WSDMS (o)** | 0.637 | 0.676 | 0.655 | 0.644 | 0.629 | **0.664** | 0.646 | **0.639** | 0.609 | 0.587 | 0.598 | **0.662** |
| **WSDMS-FC (o)** | **0.639** | **0.679** | **0.658** | **0.650** | **0.633** | **0.664** | **0.648** | **0.639** | **0.610** | **0.590** | **0.600** | **0.662** |

Table 3: Minformaing sentence detection results.

reserves vital information for spotting misinforming sentences efficiently.

- WSDMS effectively enhances sentence-level performance by utilizing publicly accessible article-level labels. To achieve comparable performance, baseline systems generally require massive fine-grained sentence-level annotations. Consequently, sentence-level prediction remains a pivotal contribution of our study.

## 5.4 Ablation Study

We ablate WSDMS based on the PolitiFact dataset by varying some component(s): 1) **w/o $\tau$**: Fully connect sentences and trees by removing $\tau$, i.e., WSDMS-FC. 2) **w/o NLL**: Replace the loss with an ordinary negative log-likelihood loss function. 3) **w/o wc**: Infer article veracity based on the original MIL assumption without weighted collective attention. 4) **Title as sent**: Treat the title as a common sentence. 5) **w/o kernel**: Reduce the kernel-based post interaction embedding to dot-product attention between sentence and conversation trees. 6) **w/o tree**: Remove conversation trees.

Figure 3 shows that most of the ablations make the result worse. **w/o tree** implies that only using article content is insufficient for the task. **w/o kernel** supports that embedding post interactions with kernel can help post and tree representation. Experiment in the Appendix A.3 also echoes the advantages of the kernel. **Title as sent** means that the news title may attract the most attention from the trees, which can hurt the representation of other sentences, and should be specially treated. **w/o wc** indicates adopting weighted collective MIL is better. **w/o NLL** confirms that our designed loss is necessary and effective. Only **w/o $\tau$** is marginally better due to fully connected sentences and trees, which is however more costly and less efficient.

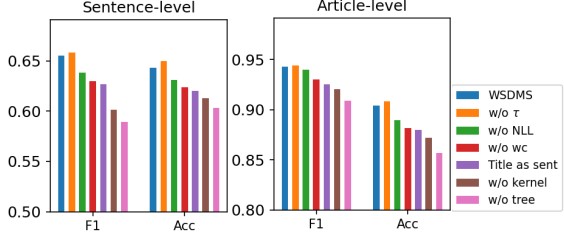

Figure 3: Ablation results on PolitiFact dataset.

## 5.5 Case Study

To gain a deeper insight, we visualize two news articles checked by PolitiFact in Figure 4 which are predicted as fake (left) and true (right) correctly by WSDMS. The spotted misinforming and true sentences are also shown. We observe that 1) WSDMS can associate a sentence with multiple trees using attention weights (arrow lines indicate high-weight trees) to help determine its veracity. 2) The posts in the conversations provide useful clues for indicating how credible each sentence is by aggregating collective opinions of users in the trees; 3) The article-level veracity is not determined simply by whether there is a misinforming sentence detected, because the prediction might be inaccurate. For example, if $s_4$ is incorrectly predicted as fake, the article will also be determined as fake under the standard MIL. Our approach increases the chance of correcting such an error by giving higher attention weights to other sentences, which may indicate that the article is overall more likely to be true. Thus, the attention weights of sentences can collectively aggregate sentence-level predictions to improve the final prediction.

## 5.6 User Study Experiment

We conduct a user study to evaluate the quality of the model output. We sample 120 articles from PolitiFact and present them in two forms: Baseline (article, posts) and WSDMS (article, misinforming

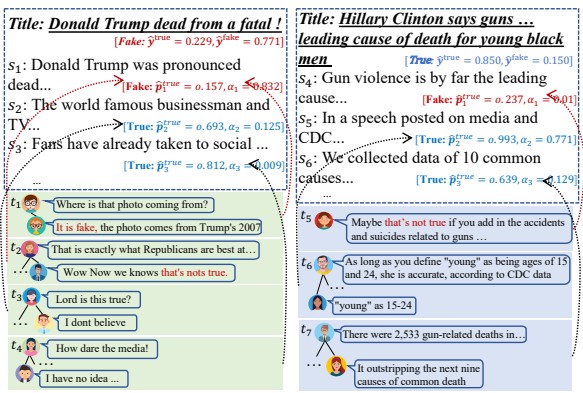

Figure 4: A case study illustrating the prediction.

|  | F1 | Acc | Confidence | Avg. Time/news |
|---|---|---|---|---|
| **Baseline** | 0.784 | 0.795 | 2.017 | 10 sec |
| **WSDMS** | 0.958 | 0.989 | 3.206 | 3 sec |

Table 4: User study results on model outputs quality.

sentences, trees). We then ask 6 users to label the articles and give their confidence in a 5-point Likert Scale (Joshi et al., 2015), and each person is given only one form to avoid cross influence.

Table 4 shows that 1) users determine the article-level veracity more accurately with WSDMS; 2) users spent 70% less time identifying fake news; and 3) users show higher confidence with the results of WSDMS, suggesting that users tend to be more sure about their decision when specific misinforming sentences and relevant evidence are provided.

## 6 Conclusion and Future Work

We propose a MIL-based model called WSDMS to debunk fake news in a finer-grained manner via weakly supervised detection of misinforming sentences with only article veracity labels for model training. WSDMS uses the attention mechanism to associate news sentences with their relevant social news conversations to identify misinforming sentences and determine the article's veracity by aggregating sentence-level predictions. WSDMS outperforms a set of strong baselines at the article level and sentence level on three datasets.

In the future, we will incorporate more inter-sentence features, such as discourse relations, to detect composition-level misinformation.

## Limitations

Fake news is one type of misinformation, which also includes disinformation, rumors, and propaganda. WSDMS can be well-generalized to detect these various forms of misinformation. Whereas, we simplify some techniques in this paper. For example, the representation of conversation trees can be learned by considering the direction of message propagation and combining top-down and bottom-up propagation trees. In addition, it cannot deal with more complex situations, where multiple true sentences combined constitute some kind of logical falsehoods or inconsistencies. This can be strengthened by considering sentence-level relations such as discourse information in the model. Despite this limitation, WSDMS encounters no such situation in the three datasets used according to our observation. Nevertheless, this suggests that the existing fake news datasets and detection models lack consideration of discourse-level fakes or logically inconsistent compositions, which are presumably not uncommon in real-world fake news. Lastly, we only use social context data collected from Twitter, which might have platform bias. To mitigate the issue, we can introduce additional data from different social media platforms, such as BuzzFace (Santia and Williams, 2018) from Facebook.

## Acknowledgements

This work was partially supported by Hong Kong RGC ECS (Ref. 22200722), National Natural Science Foundation of China Young Scientists Fund(No. 62206233) and HKBU One-off Tier 2 Start-up Grant (Ref. RCOFSGT2/20-21/SCI/004).

## Ethics Statement

**Data Privacy**: Although the datasets used in our research are publicly accessible, the utilization of social media conversations for debunking fake news may raise concerns regarding user privacy. To address this issue, we took measures to anonymize all social media posts during the data processing and experiments, ensuring that user information remains invisible and unusable. Additionally, our proposed approach does not require access to any sensitive user information, therefore eliminating the risk of privacy infringement. The collection of social media conversations in the BuzzNews dataset was conducted in compliance with the privacy regulations set by the platform.

**Social Implications**: The detection and debunking of fake news can carry significant social and political implications. One critical consideration is the potential impact on the reliability of the system and the possibility of misleading users by mislabeling

information as misinformation or vice versa. In light of this concern, we have taken precautions to carefully assess the model we developed and restrict their distribution to the general public. We are committed to designing a responsible policy regarding the dissemination of codes and datasets within research community, and ensure that they are used responsibly in a manner that aligns with ethical standards and societal well-being.

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

# A  Appendix

## A.1  Detailed Baseline Settings

Existing fake news detection and rumor detection methods predominately focus on coarse-level classification on the entire article and claim, respectively, while our goals include identifying misinforming sentences within an article at a fine-grained level. When comparing with the baselines that are originally designed to either classify a news article or a claim, the required (and available) inputs may differ from our study. Therefore, we need to specifically customize the data inputs to make the baselines applicable to the article-level and sentence-level detection tasks while maintaining the implementation of baseline models intact. In this section, we will provide more details about baseline models and the information they used.

### A.1.1  Article-level Task

1) **DeClarE** (Popat et al., 2018) is designed to classify a claim with relevant news content obtained from external sources as evidence, such as web search results. The claim it used is short and there are many relevant articles providing evidence. In our fake news detection dataset, however, what is available includes a single long-form article which is the target to be checked, and the relevant social conversation trees providing external assistance. Since DeClarE can only accept short claims as input, we use the title of the news article as an input claim and the posts in conversations as evidence.

2) **HAN** (Ma et al., 2019) aims similarly to DeClarE to the claim verification task and the provided evidence set is collected from multiple documents relevant to the claim. In our case, article text is the target to be verified, while HAN assumes a short claim as the target which cannot be fed into HAN directly. So, we use the news title as the input claim and posts in conversations as evidence.

3) **dEFEND** (Shu et al., 2019a) is a fake news detection model using news article as the target of verification and the related user comments as evidence. This is mostly consistent with our setting. Thus, it does not require any special treatment.

4) **BerTweet** (Nguyen et al., 2020) is a pretrained language model trained on large English posts corpus. It is designed to encode short text. To apply BerTweet for article-level verification, we use the posts in conversation trees to fine-tune the model, and then treat the news title as a claim to be verified because BerTweet cannot accept the

| Method | Article-level | Sentence-level |
|---|---|---|
| **DeClarE** | Title as news content. Posts as evidence. | Sentence as claim. Linked posts as evidence. |
| **HAN** | Title as news content. Posts as evidence. | Sentence as claim. Linked posts as evidence. |
| **dEFEND** | Article as news content. Posts as evidence. | Sentence as news content. Linked posts as evidence. |
| **BerTweet** | Title as news content. Posts fine-tune model. | Sentence as claim. Linked posts fine-tune the model. |
| **GCAN** | Title as news content. Users of posts as evidence. | Sentence as claim. Users of linked posts as evidence. |
| **Bi-GCN** | Title as news content. Posts as evidence | Sentence as claim. Linked posts as evidence. |
| **KAN** | Article as news content. Entities from articles and posts. | Sentence as claim. Entities from sentence and linked posts. |
| **SureFact** | Title as news content. Posts as evidence. | – |

Table 5: Application of baselines to suit the fake news datasets while keeping their original implementation intact.

long-form article as input.

5) **GCAN** (Lu and Li, 2020) aims at debunking rumors only using the corresponding sequence of retweet users without text comments of a source tweet. The source tweet it accepts as a claim is also short. To apply it to our data, we use the news title as source tweet and the post user profiles and propagation structure without post content as evidence.

6) **Bi-GCN** (Bian et al., 2020) utilizes bi-directional Graph Convolutional Network to accommodate top-down and bottom-up post propagation structure to detect rumors taking a short source post as input. Similarly, we use news title as a source post and post propagation structure as evidence.

7) **KAN** (Dun et al., 2021) detects fake news by identifying entity mentions in news contents and align them with the entities in the knowledge graph, which are used to learn news-entity co-attentions for better representing news text. While there are news articles in our data, we have only related posts from social media but no knowledge graph. For this issue, we use the social conversions of the article as the source to extract entities as entity contexts of the entities in the article.

8) **SureFact** (Yang et al., 2022b) groups related posts based on specific topics extracted from news content to implicitly connect news and social media content for fake news detection. It can be directly applied to our datasets.

### A.1.2 Sentence-level Task

For misinforming sentence detection, the baselines are deployed by treating a sentence in article as a claim or source post and the conversation trees linked to the sentence (see Section 4.2) as the source of evidence. In such a setting, most of the baselines can be applied to this sentence-level task in a more straightforward manner. See Table 5 for specific details.

### A.2 Implementation Details

Our model parameters are updated by back-propagation (Collobert et al., 2011) with Adam (Kingma and Ba, 2015) optimizer. We set the maximum epoch to 100, the dimension of embeddings to 512 for sentences and posts, and empirically initialize the learning rate as 0.001, and the hyperparameter $\lambda$ is set to 0.5 which is validated on a small hold-out dataset.

As for Gaussian kernels in Equation 2, we set $K = 10$. Here one kernel with parameter $\mu_k = 1$ and $\sigma_k = 0.001$ is designed for exact matching (Dai et al., 2018). The other kernels' parameter $\sigma_k = 0.01$, and their parameter $\mu_k$ is distributed within [-1, 1] evenly.

The training process is controlled to end when the loss value converges or the maximum epoch number is met.

### A.3 Experiment on Kernel Attention Concentration

We conduct an experiment to compute the entropy values of kernel attention weights used in WSDMS and compare it with dot-product attention used in GCAN, to reflect whether the learned attention weights are more focused or scattered. The lower the entropy, the more focused the attention mechanism (Clark et al., 2019). The entropy results are given in Table 6.

| | kernel | dot-product |
|---|---|---|
| **Attention Entropy** | 5.11 | 6.03 |

Table 6: Entropy score of kernel attention and dot-product attention.

We find that kernel attention bears a smaller entropy than the dot-product attention. It suggests that kernel attention has a stronger ability to be focused on a few more vital posts. This is also the reason why we use kernel attention in our method.