# OpenReview forum: "WSDMS: Debunk Fake News via Weakly Supervised Detection of Misinforming Sentences with Contextualized Social Wisdom"
_EMNLP/2023/Conference — EMNLP 2023 Main_

### Official Review · Reviewer_np1j · 2023-08-02

**Soundness:** 4

**Excitement:**

4: Strong: This paper deepens the understanding of some phenomenon or lowers the barriers to an existing research direction.

**Justification For Ethical Concerns:**

While not requiring an ethics review, I find the statement about not publicly releasing the model due to ethical reasons misplaced (650-656). Such restrictive practices harm reproducibility and trust in the system. Additionally, an open model would enable checking the reliability and verifying (mis-)labeling. Hence, the authors seem to act opposite to their stated ethical concerns.

**Missing References:**

027-036: The motivation of the problem is weak and lacks references to support the claims. The paper only lists a single article as an example.

Below, I suggest two papers that tie in neatly to strengthen the motivation. The first provides evidence that COVID-19 misinformation has increased virality. The second highlights the challenges of automatic fact-checking.

Solovev, K., & Pröllochs, N. (2022, April). Moral emotions shape the virality of COVID-19 misinformation on social media. In Proceedings of the ACM web conference 2022 (pp. 3706-3717).
Glockner, M., Hou, Y., & Gurevych, I. (2022). Missing counter-evidence renders nlp fact-checking unrealistic for misinformation. arXiv preprint arXiv:2210.13865.

**Paper Topic And Main Contributions:**

The paper introduces a multiple-instance-learning-based WSDMS model, which (similar to rumor detection) uses social conversations to support fake news detection. WSDMS predicts sentence veracity, which (together with attention weights) informs the article veracity prediction. The experiments show that the proposed approach leads to improved fake news prediction at the article level, while demonstrating the benefits of the sentence-level prediction task. The contribution is timely and relevant for EMNLP.

**Questions For The Authors:**

Question A: Why is there no emphasis on fact-checking? The user study is also never mentioned in the main body. In my opinion, this would be a strong direction for future research.

**Reasons To Accept:**

S1 - Approach: Considering social conversations for sentence-level predictions is interesting, feasibly conducted, and with promising results.
S2 - Experiments: The range of experiments (including choice of datasets and baselines) logically follows the overall contribution.
S3 - Case/User Study: The case and the Appendix-only user study also demonstrate the benefits of the model. Moreover, I encourage the authors to include a brief discussion of the user study in the main paper.

**Reasons To Reject:**

W1 - Sentence-level-only Performance: The sentence-level performance evaluation in Table 3 is (as noted by the authors) unfair. WSDMS is mediocre due to its disadvantage in the task (no sentence labels). WSDMS (o), on the other hand, has an advantage (pre-trained on train set). Hence, I find it troublesome to emphasize sentence prediction as prominently in the title and abstract.
W2 - Reproducibility: If possible, I expect releasing the novel datasets and models. Also, see my comment in the ethical section.

**Reproducibility:**

3: Could reproduce the results with some difficulty. The settings of parameters are underspecified or subjectively determined; the training/evaluation data are not widely available.

**Reviewer Confidence:**

4: Quite sure. I tried to check the important points carefully. It's unlikely, though conceivable, that I missed something that should affect my ratings.

**Typos Grammar Style And Presentation Improvements:**

Inconsistencies:
- threads vs. trees
- 076: Referring to t_2 is confusing here, as Figure 1 only points to t_1 and t_3.
- 238:-243: different naming than Figure 2.

Typos:
- 379: s_i should be s^hat_i?
- Figure 2: WSDM (missing an S).

Other writing issues:
- 345: Briefly clarify non-standard symbols like ⊕. Here, it likely refers to a direct sum.
- 414: Provide more details on the guide used by the annotators.
- Table 2, 457, 464: Briefly explain how groups 1 and 2 differ. Non-structured vs. structured?

---

> ### Author Rebuttal · Authors · 2023-08-28
>
> Thank you for your helpful review and feedback!
>
> 1.	W1 – Sentence-level-only Performance. Let us clarify the rationale behind the proposed WSDMS model: No existing fake news detection dataset provides labels of misinforming sentences, while all sentence-level baseline models require sentence labels for training. Given the tremendous costs of locating and annotating sentence-level misinformation in news articles, it is impractical to scale up such sentence-labeled datasets (e.g., to the size of the original training dataset), rendering the baseline models unable to get fully trained. WSDMS inherently is not bound by this limitation as it doesn’t require sentence-level annotation to predict sentence labels. Therefore, it can enjoy a much larger training set of new articles (with article-level annotations only) to boost its sentence-level prediction performance like what WSDMS (o) has demonstrated, while the baselines models can’t do this at all with the same training data. We believe this is a unique advantage of our model and it is fair to demonstrate the advantage in this way.
>
>
> 	Basically, we want to convey that WSDMS can boost sentence-level performance by training on abundantly available article-level labels, but the baselines can’t. For baselines to get comparable performance, there is an impractical high bar to prepare fine-grained sentence-level annotations. Therefore, sentence prediction is one of our main contributions.
>
>
> 	As for the WSDMS trained on the smaller test-training data with sentence annotations, which didn’t show an advantage compared with baselines, it is not surprising since its capability is intentionally weakened in such setting given a very limited number of article labels available (147/168/53). We think this doesn’t detriment the strength of WSDMS. If predominant emphasis of sentence prediction in the title and abstract still makes people feel uncomfortable, we will adjust our tone and add some specific discussions following the text in Section 5.3.
>
> 2.	W2 - Reproducibility: We will release the dataset publicly. For the source code, we will consider sharing it more responsibly. Rather than directly posting codes on a public webpage, we plan to only post the detailed usage and performance together with a request form for people who are interested in our code to log their contact and explain why they need the code and for what purpose. Then, we will verify each request via direct communication with the requesters to ensure that it will be used for research purposes only, which will be liable for a legal agreement. We believe this will be compatible with our ethical statement and do not harm the reproducibility of our method by other researchers.
>
> 3.	Question A - No emphasis on fact-checking: Fact-checking is a related area. We commented it out after we quickly balanced the space limit and the fact that our problem setting, datasets, and the angle of approaching to the misinformation challenge are different. Typically, fact-checking aims to verify a given claim rather than a long-form news article. Also, our method needs to locate and predict the misinforming sentences within the article which is also not considered in existing fact-checking tasks. We will add the related content back in the revised paper. For the user study, we moved it to the appendix completely while forgetting to leave a mention of it in the main text. We will put it back to the main text and discuss its value for future research. Since one more page of content will be allocated in the camera-ready paper, we will definitely have enough space to contain these discussions if accepted.
>
> 4.	We will correct the typos and add the reference papers accordingly to the revised paper. Thanks!

---

### Official Review · Reviewer_HG47 · 2023-08-04

**Soundness:** 4

**Excitement:**

4: Strong: This paper deepens the understanding of some phenomenon or lowers the barriers to an existing research direction.

**Paper Topic And Main Contributions:**

The paper provides an approach to sentence-level and article-level misinformation detection using weak supervision.
The authors leverage many findings of existing literature to devise their strategy, which is appreciated.
The paper is clear, well written, and present results that surpass SOA on the provided datasets.
The architectural and experimental section appear to be sound.

**Questions For The Authors:**

- Why did you chose BERT over other methods that could be more effective in modelling sentence-level representations?
- In your paper you use the term misinformation a bit as an umbrella term. In the literature, there is a small but relevant difference in meaning between misinformation and disinformation, pertaining the intentionality (i.e., misinformation is mostly unintentional, while disinformation is intentional). I understand the use of the term in your context, but I believe that it could be worth it to clarify this aspect in the paper.

**Reasons To Accept:**

The paper faces a challenging problem in a creative but also effective way. While the approach leverages many existing techniques, this is not a shortcoming, as authors are able to incorporate them in their own method.
While not a definitive answer to the problem of misinformation, I believe that it may be an interesting stepping stone of the branch of the literature focusing on social contexts.

**Reasons To Reject:**

The paper leverages few seemingly outdated or out-of-scope approaches, e.g. a base BERT model for sentence-level representations, which is far from optimal. Maybe a different model like Sentence-BERT would have performed better for the same computational cost.
Also, the experiments are conducted on three well know and generally easy datasets. In my opinion, experiments on more challenging datasets (e.g., those were fake news are orders of magnitude less than real ones, as in real-world scenarios) are better suited for actually improving real-world fake news detection.

**Reproducibility:**

4: Could mostly reproduce the results, but there may be some variation because of sample variance or minor variations in their interpretation of the protocol or method.

**Reviewer Confidence:**

4: Quite sure. I tried to check the important points carefully. It's unlikely, though conceivable, that I missed something that should affect my ratings.

**Typos Grammar Style And Presentation Improvements:**

The paper is well written. My only minor gripe is the one mentioned above concerning the term misinformation, that I think could be changed or clarified to reflect the intentions of the authors.

---

> ### Author Rebuttal · Authors · 2023-08-28
>
> We appreciate your insightful feedback and suggestions! Essentially, these are very similar concerns to the Reviewer BSaA.
>
> 1.	As we explained above, we used BERT to have a fairer comparison with the baselines, such as BerTweet, KAN, and SureFact which also used base BERT. We are committed to including experimental results using Sentence-BERT in our revised paper.
>
> 2.	We agree that fake news detection requires more challenging dataset. The datasets we used present certain degree of class imbalance as shown in Table 1, where the number of fake news accounts for 30-50% of that of true news. We concur that this might not a reflection of the real-world ratio, but the difficulty is we do not know what the real-world ratio is, and we’re not aware of any existing dataset reflecting a real-world ratio. We think it is a very interesting research question to answer in the future. Again, if you know any references regarding the real-world fake news ratio or any more realistic dataset, we’ll be very thankful and determined to test out our approach on that!
>
> 3.	Thanks again for your helpful suggestion on the use of umbrella term “misinformation”. Indeed, we didn’t differentiate misinformation and disinformation strictly here even though we’re aware of their difference. We will clarify it and define disinformation to be a subset of misinformation that our approach can generally deal with in the revised paper.

---

### Official Review · Reviewer_BSaA · 2023-08-04

**Soundness:** 3

**Excitement:**

4: Strong: This paper deepens the understanding of some phenomenon or lowers the barriers to an existing research direction.

**Paper Topic And Main Contributions:**

This paper presents a model called Weakly Supervised Detection of Misinforming Sentences (WSDMS) to detect sentence-level misinformation and article-level veracity and evaluates the performance of the proposed model on three real-world benchmarks. Drawing from existing literature, the well-articulated research showcases results that eclipse the state-of-the-art (SOA) benchmarks. The model adeptly predicts sentence veracity and, combined with attention weights, guides the article's overall truthfulness assessment, proving instrumental in enhancing fake news detection while underscoring the significance of sentence-level prediction.



**Reasons To Accept:**

This paper presents a model to address a challenging problem, detecting sentence-level misinformation and article-level veracity, and evaluates the performance of the proposed model on three real-world benchmarks. The literature is critically presented, and the authors considered social conversations for sentence-level predictions. The overall contribution of the study is shown by the depth of the experiments.



**Reasons To Reject:**

This paper, while intriguing and worthy of consideration for publication, employs certain outdated methods like the base BERT model for sentence representations, which may not be optimal compared to models like Sentence-BERT. The chosen datasets for experimentation are widely known and relatively straightforward; testing on more challenging datasets, where fake news is sparse as in real-world situations, would provide a more genuine insight into its applicability. Since fake news detection is a hard problem, so complex datasets should be used to address this task.

**Reproducibility:**

3: Could reproduce the results with some difficulty. The settings of parameters are underspecified or subjectively determined; the training/evaluation data are not widely available.

**Reviewer Confidence:**

5: Positive that my evaluation is correct. I read the paper very carefully and I am very familiar with related work.

---

> ### Author Rebuttal · Authors · 2023-08-28
>
> Thanks a lot for your great effort and positive feedback on our submission!
>
> 1.	It is true that the BERT base model can be replaced with other state-of-the-art representation methods such as Sentence-BERT. The main reason we just used BERT is to have a fairer comparison with the baselines such as BerTweet, KAN and SureFact, all of which employed BERT for sentence representations. We ensure to include Sentence-BERT-based experimental result into our revised paper.
>
> 2.	We agree that fake news detection is a hard problem requiring more challenging dataset. To the best of our knowledge, the three datasets we used are representative and widely known. Besides, the datasets have already considered imbalanced classes where the number of fake instances is around 30-50% of the number of true instances, as shown in Table 1. We conjecture that the datasets creators could further reduce the ratio to make the fake class even sparser, but the difficulty might be how to know what the realistic (or gold) ratio is in real-world situations. It is a very interesting research question that we are keen to find out in the future, and we’re not aware of any existing fake news datasets representing or approximating such a gold ratio of fake vs. true. If you know any references regarding the real-world fake news ratio or any more realistic dataset, we’ll be very thankful and determined to test out our approach on that.

---

### Meta-Review · Area_Chair_J7PF · 2023-09-18

**Recommendation:** 4

**Metareview:**

This paper introduces the task of identifying sentence-level misinformation, as part of fake-news debunking. Due to the lack of an appropriate dataset, multi-instance learning is used as a form of weak supervision, and achieves state of the art results.

Overall, reviewers (all of whom were quite confident) were Impressed by both the writing ("well-articulated research") and the research ("faces a challenging problem in a creative but also effective way").

Two reviewers raised nearly identical concerns -- the use of an outdated model as the basis for the method, when a newer approach (SentenceBERT) might have performed better, and the use of datasets that are known to be relatively easy. The authors have committed to including results with the first suggestion in their final version, but note that they are limited by the lack of a more suitable dataset for the second.

Although these do seem like shortcomings, reviewers were nearly unanimous in this paper being strong on both soundness and excitement. As such, this seems like a clear accept ("the contribution is timely and relevant"). None were particularly effusive in their written praise (e.g., "intriguing", and an "interesting stepping stone", rather than transformative), but all rated it as strong on excitement, and wish to see it published.

---

### Decision · Program_Chairs · 2023-10-07

**Decision:**

Accept-Main

**Comment:**

This paper introduces the task of identifying sentence-level misinformation, as part of fake-news debunking. Due to the lack of an appropriate dataset, multi-instance learning is used as a form of weak supervision, and achieves state of the art results.

Overall, reviewers (all of whom were quite confident) were Impressed by both the writing ("well-articulated research") and the research ("faces a challenging problem in a creative but also effective way").

Two reviewers raised nearly identical concerns -- the use of an outdated model as the basis for the method, when a newer approach (SentenceBERT) might have performed better, and the use of datasets that are known to be relatively easy. The authors have committed to including results with the first suggestion in their final version, but note that they are limited by the lack of a more suitable dataset for the second.

Although these do seem like shortcomings, reviewers were nearly unanimous in this paper being strong on both soundness and excitement. As such, this seems like a clear accept ("the contribution is timely and relevant"). None were particularly effusive in their written praise (e.g., "intriguing", and an "interesting stepping stone", rather than transformative), but all rated it as strong on excitement, and wish to see it published.